# The influence of glottal and respiratory factors on aerosol emission during phonation

Allison Hilger[1*], Tehya Stockman[2¤a], Corey Murphey[3], Jacqueline McCurdy[1¤b], Shelly Miller[2]

1 Department of Speech, Language, and Hearing Science, University of Colorado Boulder, Boulder, Colorado, United States of America, 2 Department of Mechanical Engineering, University of Colorado Boulder, Boulder, Colorado, United States of America, 3 Department of Computer Science, University of Colorado Boulder, Boulder, Colorado, United States of America

¤a Current address: Adams County, 4430 S Adams County Pkwy, Brighton, United States of America
¤b Current address: HCA HealthONE Sky Ridge Outpatient Rehabilitation, Colorado, United States of America
* allison.hilger@colorado.edu

## Abstract

### Introduction

Speech-driven aerosol generation plays a key role in airborne disease transmission, yet the physiological mechanisms remain poorly understood. Beyond vocal-fold vibration, airflow and glottal configuration may be key determinants. We tested how phonation type affects aerosol generation while accounting for ventilatory output estimated from exhaled $CO_2$.

### Methods

Five healthy female adults (22–43 years) sustained vowels across six phonation types: modal register, glottal fry, falsetto register, forced whisper, loud modal register, and vowels preceded by/h/. Aerosol concentration and size distribution (0.1–20 µm) were measured using an aerodynamic particle sizer (APS). Laryngoscopy, conducted in a separate session, was used to estimate the normalized glottal gap during the open phase of phonation (NGG). Exhaled $CO_2$ range was recorded concurrently as a control for ventilatory variation across tasks.

### Results

Phonation types that had greater ventilatory output and a larger open-phase glottal gap (e.g., forced whispering, loud modal register) produced the highest aerosol concentrations; types with less ventilatory output and smaller open-phase glottal gap (e.g., glottal fry and modal register) produced the lowest. Submicron particles (0.1–1 µm) dominated across conditions. Forced whispering exhibited a bimodal aerosol distribution, with increased emissions at both the smallest (0.1–1 µm) and largest

---

**Data availability statement:** The minimal dataset needed to replicate the findings has been deposited at figshare: doi.org/10.6084/m9.figshare.30251644.

**Funding:** This study was made possible with funding from and partnership with the National Federation of State High School Associations and the College Band Directors National Association, as well as the Research and Innovation Office at the University of Colorado Boulder. The funders had no role in study design, data collection and analysis, decision to publish, or preparation of the manuscript.

**Competing interests:** The authors declare that they have no competing interests; this does not alter our adherence to PLOS ONE policies on sharing data and materials.

(10–20 μm) particle sizes. Despite the assumption that vocal fold vibration is necessary for aerosol production, whispering, a voiceless sound production, generated a high concentration of particles, suggesting a primary role for airflow and glottal configuration. Normalized glottal gap was the strongest predictor of aerosol output, and $CO_2$ range (ventilatory output) was also positively associated.

## Conclusion

Sustained sound production can generate substantial aerosols even without vocal fold vibration. The strong association between normalized glottal gap and aerosol output indicates that airflow and glottal configuration, rather than vibration alone, are primary contributors under these task conditions.

---

## Introduction

The rapid airborne transmission of SARS-CoV-2 during the COVID-19 pandemic emphasized the importance of understanding how speech and singing contribute to aerosol dispersion, particularly in enclosed spaces such as restaurants, classrooms, and choir settings [1–3]. Airborne pathogens such as COVID-19, influenza, and tuberculosis spread through exhaled aerosol particles generated during breathing, coughing, sneezing, and vocalizing (i.e., phonation from speaking or singing) [4–6]. Despite the established role of exhaled particles in disease transmission, the specific physiological mechanisms underlying aerosol generation in the respiratory and vocal tracts remain poorly understood. Here we refer to an aerosol as particles suspended in a gas, and an airborne droplet as a small particle of liquid often containing a virus or bacteria that is released when someone talks or sings. Droplets evaporate rapidly upon emission, and the diameter of the droplet shrinks, but this phenomenon is not well understood [7].

When comparing expiratory events, speech and singing generate more particles per unit of time than coughing or breathing, with smaller particles that remain airborne longer and increase the likelihood of inhalation by others [8,9]. Louder speech has been shown to generate higher aerosol concentrations than quiet speech, raising the question of whether this effect is driven by increased lung pressure, greater amplitude of vocal fold vibration, or both [8,10]. Recent work reports that whispering can produce higher aerosol concentrations than the modal register [11,12]. Although mechanisms were not measured, whisper tasks typically involve a wider glottal configuration and can be produced with substantial ventilatory output. Taken together, these findings challenge the assumption that quieter vocalization necessarily lowers airborne-transmission risk.

Theories of aerosol generation propose that exhaled particles originate from multiple points within the respiratory tract. In the lungs, a fluid film across the bronchioles bursts during exhalation, generating small particles (<1 μm) [9,10,13]. In the larynx, particles can form due to vortex shedding and vibration-induced atomization caused by the vibration of the vocal folds (~0.5–5 μm) [5]. In the oral cavity, the motion and

approximation of the oral articulators, including the lips, tongue against the teeth and palate, and the velopharyngeal port, create multiple opportunities for particle formation via fluid-film rupture and oral jetting, which generate particles that tend to be larger (~5–20 μm) [9,14–16]. These size distinctions are significant, as smaller laryngeal-generated particles (~0.5–5 μm) remain airborne longer and fit within the size range required for conveying many viral and bacterial particles, such as tuberculosis bacilli, COVID-19, and influenza [17–19].

Changes in phonation quality and ventilatory output significantly affect aerosol concentration and emission rates (and possibly particle size). Loud speech and singing typically require greater subglottal pressure, often initiated by higher lung volumes (~60–90%VC), which may increase aerosol generation [8,20–22]. Voiced sounds (e.g.,/b, g, z/) generate more aerosol particles than voiceless sounds (e.g.,/p, k, s/), suggesting a role for vocal fold vibration in aerosol production [23]. Emissions are also sensitive to ventilatory pattern: higher expiratory flow and exhalation below functional residual capacity are associated with increased particle counts, whereas breath-hold (no flow) yields minimal emission [10,24]. Further, whispering and loud speech generate high aerosol concentrations, reinforcing the importance of airflow dynamics in vocalized aerosol generation [8,11,12,25].

Despite increasing research interest in aerosol generation from speech, critical gaps remain in understanding how phonatory biomechanics and respiratory control interact to influence aerosol output. Many studies have focused on speech intensity and phoneme type, but few have systematically controlled ventilatory output, glottal positioning, or phonatory condition. This study aims to determine which physiological factors most strongly predict aerosol generation during phonation. Specifically, we investigate the relationship between carbon dioxide range ($CO_2$ range) and maximal glottal gap during the open phase of phonation across phonatory conditions to assess their impact on particle concentration. Each of these measures reflects distinct physiological processes that can influence aerosol generation during vocalization. $CO_2$ range serves as an indicator of ventilatory output, which could be reflective of lung volume; both of these measures can affect the velocity and force of exhaled airflow, thereby influencing the number and size of emitted particles [26,27]. We use ventilatory output to mean the relative amount of exhaled gas per unit time during a trial, indexed by a capnography-derived $CO_2$ range measure. This covariate serves only as a control for between-trial differences in ventilation; it is not a direct measure of airflow, subglottal pressure, muscle activation, or absolute lung volume. The glottal gap reflects the degree of maximal vocal fold separation during the open phase of phonation (obtained using laryngoscopy in a separate session), which affects the pressure and airflow dynamics as air passes through the glottis, potentially impacting aerosol formation. By comparing these measures, the study aimed to identify the primary physiological mechanisms driving aerosol production during vocalization.

To achieve this, we had study participants produce six distinct phonation conditions (modal register, glottal fry, falsetto register, forced whispering, loud modal register, and the/h/ sound prior to a sustained vowel) to evaluate how variations in ventilatory output and glottal positioning affect aerosol output. Phonation types vary in their acoustic output and underlying physiological mechanisms, which may influence aerosol generation. Across phonatory conditions, we refer to phase proportions of the vocal fold cycle rather than categorical closure. Specifically, closure quotient (CQ) denotes the fraction of each cycle in which the folds are closed.

Modal register is characterized by a stable, periodic waveform with an intermediate closure quotient (CQ); both open and closed phases are well represented within each vibratory cycle [28]. The second phonation quality is glottal fry, which occurs at very low fundamental frequencies and typically shows a higher CQ, with prolonged closed phases and brief, burst-like openings [29]. The falsetto register is produced with thin, elongated folds, a higher fundamental frequency, and a lower CQ, often with a predominantly open phase and, in some speakers, incomplete closure [30]. The fourth phonation quality is forced whispering, which is voiceless: the folds do not vibrate; instead, an abducted configuration produces aperiodic turbulence [31]. Fifth is a loud modal register in which a clear voice is used, but at 10 dB SPL louder than comfortable voicing in this study. Finally, the last phonation quality is saying/h/before each vowel, for example, "heeeee" or "haaaaa." This final voicing style is produced with a partially open glottis for the/h/ sound, followed by modal phonation.

These six conditions span differences in register (fry, modal, falsetto), voicing state (voiced vs. whispered), and intensity (comfortable vs. loud), allowing us to examine how glottal configuration and ventilatory output relate to particle concentration and size without asserting a single causal mechanism. We anticipated higher ventilatory output (indexed by $CO_2$ range) for loud modal phonation and whispering relative to the other tasks [32], providing a principled contrast set for testing our hypotheses about aerosol production during sustained phonation.

## Methods

### Participants

This experiment included five healthy female participants, all between the ages of 22–43 years (mean = 31), with no history of voice or respiratory impairments. The study was approved by the University of Colorado Boulder Institutional Review Board (IRB Protocol #20–0282) for the aerosol measurements and the Colorado Multiple Institutional Review Board (COMIRB Protocol #22–1678) for the laryngoscopy. Participants were provided with informed written consent. Participant recruitment for IRB Protocol #20–0282 occurred from 01/01/2022–08/16/2022. Follow-up laryngoscopy occurred on December 8th, 2022, under COMIRB Protocol #22–1678.

### Procedures

To measure particle concentration and size across different phonation types, participants produced sustained vowels using six distinct phonation qualities: modal register, glottal fry, falsetto register, forced whispering, loud modal voicing, and vowels preceded by an /h/ (i.e., h-onset). Each phonation quality was produced across four sustained vowels, /ɑ/ ("saw"), /i/ ("see"), /ɛ/ ("said"), and /ʌ/ ("sun") chosen for their differences in tongue position. The goal was to measure aerosol generation across phonation tasks varying by open vocal tract postures to maximize generalizability. Note that for forced whispering, participants produced a sustained vowel in a voiceless whisper. This instruction elicits a high-intensity "stage whisper." No phonation (vocal fold vibration) was intended.

Participants sustained each vowel for five seconds and repeated the production three consecutive times per phonation condition. To ensure consistency, a timed visual presentation guided participants through the sequence of vowel productions and phonation types. Participants vocalized into a funnel connected to an aerodynamic particle sizer (APS, 3321, TSI) to measure particle size distributions for particles that reach the APS sensor at the end of the funnel. The APS measured particle size distributions of ~0.1–20 μm. Additionally, $CO_2$ concentrations were measured using a Licor (LI-7000, Li-Cor, Lincoln, NE), which samples $CO_2$ once per second, with the data averaged every minute. The APS measured a size distribution every minute. To reduce background aerosol concentration, two portable HEPA air cleaners (Air Response Air Purifier, Oreck) were used to decrease the background levels of aerosol between runs. The total particle number background concentration in the room was 0.03–0.1#/cm3 as reported by the APS.

Participants wore an AKG C420 condenser microphone positioned over the ear, connected to a Motu Ultralite-mk3 Hybrid audio interface. A real-time visual feedback system displayed their voice loudness, helping them maintain a target range of 70–75 dB SPL during phonation. Each sequence of three vowel productions was followed by 35 seconds of nasal breathing, allowing sufficient time for the aerosol plume to dissipate and preventing signal overlap between trials.

In a separate testing session at an otolaryngology clinic, laryngoscopy was used to measure glottal positioning across the six phonation qualities using the normalized glottal area during the open phase of vocal fold oscillation. Nasolaryngoscopy videos were obtained from all five participants while they sustained the /i/ vowel across all six phonation qualities. A speech-language pathologist conducted the laryngoscopy using a flexible laryngoscope (Olympus, Olympus Surgical, Center Valley, PA) inserted transnasally and positioned near the arytenoid cartilages. Another speech-language pathologist captured still images of the open phase of oscillation for three frames per phonation type.

The normalized glottal gap was then measured for each still shot using ImageJ (National Institutes of Health, Bethesda, MD, USA, https://imagej.nih.gov/ij) based on the procedures used in [33]. From stroboscopic sequences, we identified the

frame of maximum vocal-fold excursion (peak open phase) and manually traced the glottal contour across three consecutive cycles; the squared membranous vocal-fold length normalized areas to yield a dimensionless normalized glottic gap (NGG). NGG was defined as the maximum interfold distance divided by vocal fold length. Vocal fold length was measured on the endoscopic image from the posterior vocal process of the arytenoid to the anterior commissure (the junction of the vocal folds at the thyroid cartilage). This length was used to set the scale function in ImageJ. The glottal area was then quantified by adjusting image brightness using the color threshold function, isolating the glottic gap. A line sector measurement was used to determine the distance at the widest point of the glottic gap. The resulting normalized glottal gap measure was calculated by dividing the anterior gap by the square of vocal fold length times 100. This process was repeated for three consecutive phonatory cycles, and values were averaged to obtain a final measurement for each participant and phonation type. For the/h/-onset condition, NGG was extracted from the vowel portion following/h/ (i.e., not during the consonantal/h/). The/h/-onset task was produced as an easy onset into a modal vowel at the target intensity.

To determine reliability, we applied the Canny edge detection algorithm [34] to a subset of 40% of the laryngoscopy images (approximately 40% of the dataset) to quantify the glottal gap. The algorithm identifies edges in digital images by computing intensity gradients and applying non-maximum suppression and hysteresis thresholding, resulting in an edge map outlining the vocal fold contours. From these edges, we computed the maximum glottal length and width through the centroid of the glottis during the open phase. Agreement between automated and manual measurements was assessed using correlation and Bland–Altman analyses to evaluate reliability. Reliability between the automated Canny edge detection algorithm and the human rater was moderate to strong, with Pearson correlation of 0.84 and Spearman correlation of 0.74. Bland–Altman analysis indicated a small mean bias (+13 units) with 95% limits of agreement spanning –827 to +853, suggesting reasonable consistency between automated and manual estimates, though some variability remained across the measurement range. Intra-rater reliability was also calculated for NGG for 16% of images and was excellent (ICC[3,1] = 0.924, 95% CI 0.853–0.969).

## Outcome measures

The primary outcome measures were particle concentration and particle size, measured as dN/dlogDp, obtained from the aerodynamic particle sizer (APS), where N is the particle number concentration and Dp is the particle diameter. To ensure a meaningful interpretation of particle size distributions, particle concentration was categorized into five size bins: 0.1–1 μm, 1–2.5 μm, 2.5–5 μm, 5–10 μm, and 10–20 μm. These bins were selected based on prior literature that distinguishes between respirable aerosols (<2.5 μm), transition-sized particles (2.5–10 μm), and larger, coarse particles (>10 μm), which have different aerodynamic behaviors and transmission risks [9,17]. Categorizing particle size enables statistical modeling to evaluate trends across biologically relevant ranges that align with infectious disease transmission models.

To define the analysis window for particle concentration and size measurements, $CO_2$ range was used as an objective criterion. Since $CO_2$ levels rise during speech, the gas analyzer provided a method to delineate speech and nasal breathing phases. The minimum and maximum $CO_2$ values were calculated for each 75-second trial window (15 seconds of speech, 60 seconds of nasal breathing). The APS analysis window was then set post facto by applying a 25% threshold of this range, ensuring that particle data captured corresponded to active phonation. This analysis window was also used to determine $CO_2$ range by subtracting the minimum $CO_2$ output from the maximum output. Specifically, from the time-capnogram recorded during each APS trial, we computed a within-trial amplitude metric (peak–trough after smoothing) and z-scored it across trials. We interpret $CO_2$ range strictly as a ventilatory output covariate and do not infer airflow, pressure, or lung volume from this signal. The other measure in this study included normalized glottal gap, described previously.

## Statistical analysis

The first research question asked how particle concentration varied by vocal task and particle size while controlling for $CO_2$. Specifically, a Bayesian regression model was implemented using the brms package [35] in R (R version 4.3.2) to

analyze the effects of vocal task and particle size on particle concentration while controlling for $CO_2$ range and including random intercepts by participant. The dependent variable was particle number concentration per diameter size range (dN/dlogDp, #/cm³), and the independent variables included vocal task (modal register, loud modal register, forced whispering, glottal fry, falsetto register, and/h/ before vowel), particle size (µm), and $CO_2$ range. The model included an interaction term between task and size to assess whether differences in particle concentration between vocal tasks varied across particle sizes. Additionally, $CO_2$ range was included as a covariate to account for the potential influence of the amount of air expelled on particle concentration. $CO_2$ range was scaled to improve model convergence and interpretability, ensuring that parameter estimates were on a comparable scale.

We modeled aerosol concentration with a lognormal likelihood, which in *brms* parameterizes the linear predictor on the log scale. Models included participant random intercepts. Predictors were z-scored. We used weakly informative priors: Normal(0, 1) on the intercept and fixed effects (log scale), and Exponential [1] on group-level SDs and on **σ**, to regularize variance components. Sampling used NUTS (4 chains × 2000 iterations; 1000 warm-up) with max_treedepth = 15 and a raised adapt_delta (≥0.95; 0.99 where needed) to avoid divergent transitions. Convergence diagnostics were satisfactory (all R̂ ≈ 1 and effective sample sizes adequate).

To evaluate the influence of the $CO_2$ range control variable, a reduced model was constructed by excluding $CO_2$ range from the predictors. The performance of the full model, which included $CO_2$ range, was compared to the reduced model using leave-one-out cross-validation (LOO). This comparison was conducted using the loo function in the R package brms [35], which calculates the expected log pointwise predictive density (elpd) for each model. The difference in elpd between the two models, along with its standard error, was used to assess whether including $CO_2$ range significantly improved the model's predictive accuracy. The results of this comparison indicated that including $CO_2$ range led to a better fit, as evidenced by a higher elpd value for the full model, supporting its inclusion as a control variable in the analysis.

We fit a second Bayesian mixed-effects model to compare the contributions of normalized glottal gap (NGG) and $CO_2$ range (ventilatory output) to aerosol concentration during phonation. Predictors were z-scored (mean = 0, SD = 1) to allow direct comparison of effect sizes. The outcome was modeled with a lognormal likelihood in *brms*, which places the linear predictor on the log scale; models included participant random intercepts. We used weakly informative priors (Normal(0,1) for the intercept and fixed effects on the log scale; Exponential [1] for group-level SDs and **σ**) and sampled with NUTS (4 chains, 2000 iterations each; 1000 warm-up; adapt_delta ≥ 0.95 and max_treedepth = 15). Coefficients were interpreted multiplicatively: exp(β) gives the ratio change in expected concentration per 1-SD increase in each predictor. Relative evidence was summarized from the posterior (medians, 95% CrIs, and the probability that |β_NGG| > |β_CO₂|).

To provide a size-agnostic summary of emissions, we ran a final Bayesian mixed-effects model on the APS-reported total number concentration per sample (#/cm³). This outcome is the instrument's total concentration for each trial and avoids any averaging across size bins. We used the same specification as above: a lognormal likelihood (linear predictor on the log scale), participant random intercepts, predictors z-scored where applicable ($CO_2$), and weakly informative priors (Normal(0,1) on the intercept and fixed effects; Exponential [1] on group-level SDs and on σ). Sampling used NUTS (4 chains × 2000 iterations; 1000 warm-up; adapt_delta ≥ 0.95, max_treedepth = 15). For inference, we obtained estimated marginal means by task with $CO_2$ fixed at its mean and computed pairwise task contrasts as ratios on the response scale using emmeans; we report posterior medians with 95% HPD intervals. This complementary analysis targets overall emission while the Task×Size model characterizes size-specific patterns.

Model convergence was assessed using the Rhat statistic, with values close to 1.00 indicating good convergence. Effective sample sizes (ESS) were examined to ensure sufficient sampling precision. Posterior distributions of model parameters were summarized using median estimates and 95% highest posterior density (HPD) intervals. Pairwise comparisons between vocal tasks were conducted using the emmeans package with Bonferroni adjustments to control for multiple comparisons. Model fit and predictive accuracy were evaluated using approximate leave-one-out cross-validation (LOO), with Pareto-smoothed importance sampling (PSIS) to assess model diagnostics. For all model results, a robust

effect was determined if the HPD interval did not include zero, indicating a statistically meaningful association between the predictor and the outcome variable. Data and code can be found at https://osf.io/uc6km/ (Fig 1).

## Results

### Overall particle concentration totals

Among vocal tasks, the loud modal register produced the highest total particle concentrations, robustly exceeding whisper (Estimate = 0.401; ratio ≈ 1.49×, 95% HPD for ratio: 1.32–1.69×), modal register (Estimate = 0.966; 2.63×, 2.34–2.94×), falsetto register (Estimate = 0.744; 2.10×, 1.89–2.35×), and glottal fry (Estimate = 1.024; 2.78×, 2.37–3.28×). Forced whispering exceeded modal register (Estimate = 0.565; 1.76×, 1.55–2.01×) and glottal fry (Estimate = 0.624; 1.87×, 1.52–2.23×). The/h/-onset vowel exceeded the modal register (Estimate = 0.621; 1.86×, 1.67–2.08×), was lower than the loud modal register (Estimate = −0.344; 0.71× of loud, 0.64–0.78×) and did not differ credibly from forced whispering.

### Particle concentration by task and size

A Bayesian contrast analysis compared aerosol concentrations across vocal tasks and particle sizes with 95% HPD intervals. The results confirm that smaller particles (0.1–1 µm) exhibit robustly higher concentrations than all other larger size ranges, including 1–2.5 µm (Estimate = 0.48, 95% HPD: [0.40, 0.54]), 2.5–5 µm (Estimate = 0.30, 95% HPD: [0.16, 0.47]), 5–10 µm (Estimate = 0.55, 95% HPD: [0.23, 0.86]), and 10–20 µm, (Estimate = 0.89, 95% HPD: [0.81, 0.97]).

A total of 222 robust interactions were identified in the post hoc pairwise comparisons, where the 95% highest posterior density (HPD) intervals did not include zero. These results indicate robust differences in aerosol concentration across vocal tasks and particle sizes. To highlight the most relevant, impactful interaction effects (shown in **Fig 2**), we have summarized the results by task and size as follows.

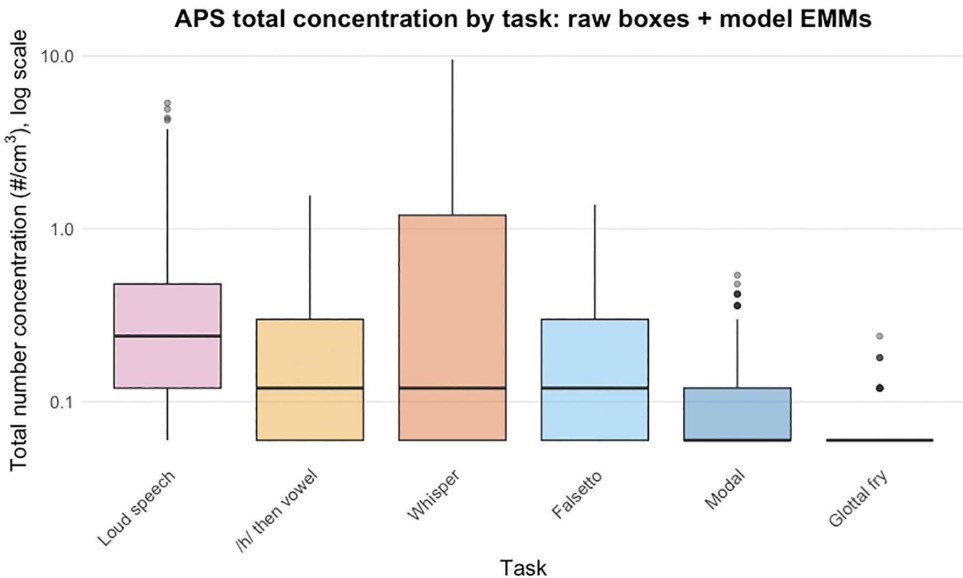

**Fig 1. Distribution of APS total number concentration (#/cm³) across vocal tasks.** The y-axis is log10-scaled (values are raw per-sample totals; no additional transformation). Boxes show the interquartile range with the median line; whiskers extend to 1.5 × IQR; points beyond the whiskers are observations outside this range. Tasks are displayed in descending order to aid comparison. Colors match task categories used throughout.

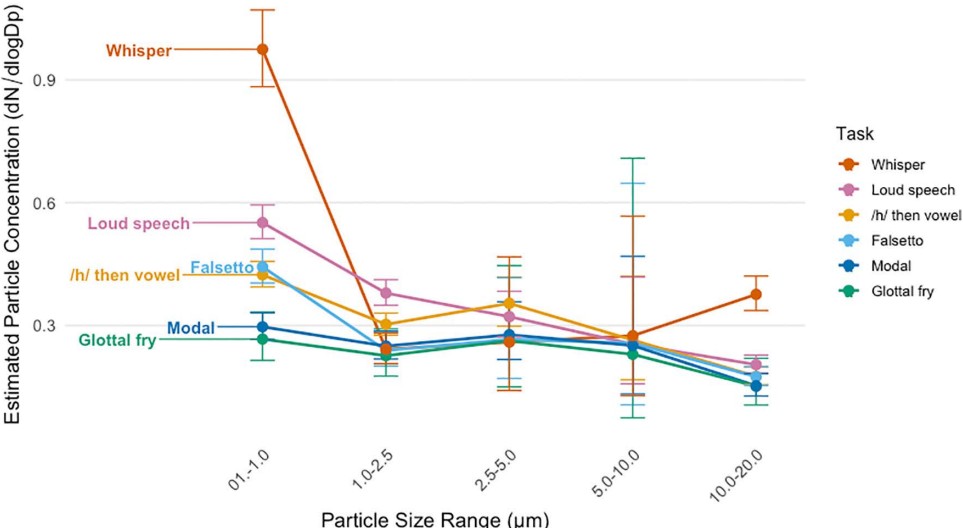

**Fig 2. Interaction between vocal task and particle size on aerosol concentration.** Estimated aerosol concentration (#/cm³) is plotted across five particle size ranges (0.1–1 µm, 1–2.5 µm, 2.5–5 µm, 5–10 µm, 10–20 µm) for each vocal task. Lines connect the estimated means for each task, with error bars representing 95% credible intervals.

## The interaction of particle concentration across particle size ranges for different vocal tasks

Pairwise comparisons of the interaction between vocal task and particle size range revealed that concentrations were highest at the smallest particle size range (0.1–1 µm) and progressively declined as particle size increased for all tasks except forced whispering. This trend was consistent across loud modal register, falsetto register, modal register,/h/ then vowel, and glottal fry, with robust differences between the smallest and largest size ranges.

**Forced whispering by particle size.** Within forced whispering, the 0.1–1 µm bin had the highest concentration, exceeding 1–2.5 µm (β = 1.58; ratio ≈ 4.9×, 95% HPD for ratio 4.17–5.55×), 2.5–5 µm (β = 1.29; 3.6×, 2.11–6.41×), 5–10 µm (β = 0.913; 2.5×, 1.18–4.94×), and 10–20 µm (β = 1.079; 2.94×, 2.61–3.29×). Notably, the 10–20 µm bin was higher than 1–2.5 µm (reported contrast 1–2.5 vs 10–20: β = −0.502; equivalently, 10–20 is 1/exp(−0.502) ≈ 1.65 × 1–2.5, 95% HPD 1.42–1.93×), indicating a secondary coarse-mode elevation unique to forced whispering in our dataset. Differences involving 10–20 vs 2.5–5, 10–20 vs 5–10, and 1–2.5 vs 5–10 were not credibly different.

**Loud modal register by particle size.** Within the loud modal register, the 0.1–1 µm bin showed the highest concentrations, exceeding 1–2.5 µm (β = 0.336; ratio ≈ 1.40×, 95% HPD 1.27–1.54×), 2.5–5 µm (β = 0.497; 1.64×, 1.40–1.93×), 5–10 µm (β = 0.714; 2.04×, 1.32–3.18×), and 10–20 µm (β = 1.097; 3.00×, 2.67–3.38×). The 1–2.5 µm bin also exceeded 10–20 µm (β = 0.763; 2.15×, 1.90–2.44×). Overall, the loud modal register exhibits a strong small-particle dominance with a marked drop toward larger sizes.

**/h/-onset by particle size.** Within/h/-onset, the 0.1–1 µm bin was highest, exceeding 1–2.5 µm (β = 0.259; ratio ≈ 1.30×, 95% HPD 1.17–1.43×), 5–10 µm (β = 0.510; 1.67×, 1.03–2.67×), and 10–20 µm (β = 0.983; 2.67×, 2.34–3.01×). The 1–2.5 µm bin also exceeded 10–20 µm (β = 0.724; 2.06×, 1.80–2.37×). Overall,/h/-onset shows pronounced small-particle dominance without the coarse-mode elevation observed for forced whispering.

**Falsetto register by particle size.** Within the falsetto register, the 0.1–1 µm bin was highest, exceeding 1–2.5 µm (β = 0.495; ratio ≈ 1.64×, 95% HPD 1.39–1.94×) and 10–20 µm (β = 0.920; 2.51×, 2.16–2.88×). The 1–2.5 µm bin also

exceeded 10–20 μm (β = 0.423; 1.53 ×, 1.26–1.83×). Overall, the falsetto register shows strong small-particle dominance with no reliable coarse-mode elevation.

**Modal register by particle size.** Within the modal register, small particles dominated primarily relative to the coarse bin. The 0.1–1 μm bin exceeded 10–20 μm (β = 0.653; ratio ≈ 1.92 ×, 95% HPD 1.61–2.33×) and 1–2.5 μm also exceeded 10–20 μm (β = 0.553; 1.74 ×, 1.42–2.12×). Overall, the modal register shows small-particle dominance with a marked suppression of the coarse (10–20 μm) range.

**Glottal Fry by particle size.** Glottal fry showed the fewest robust differences but followed the same overall trend, with higher particle concentrations in the 0.1–1 μm range compared to 10–20 μm (β = 0.612, 95% HPD: [0.230, 0.991]). Likewise, 1–2.5 μm exceeded 10–20 μm (β = 0.525, 95% HPD: [0.111, 0.928]).

Overall, these findings indicate that most vocal tasks predominantly generate small particles, with the highest concentration observed in the 0.1–1 μm range and progressively fewer emissions at larger particle sizes. However, forced whispering uniquely produces substantial emissions across both small and large particle ranges.

**Differences among vocal tasks within each particle size range**

**Smallest size bin (0.1–1 μm).** Forced whispering produced the highest particle concentration, exceeding/h/-onset (β = 0.556, ratio ≈ 1.74 ×, 95% HPD for ratio 1.56–1.96×), the falsetto register (β = 0.696, 2.01 ×, 1.78–2.26×), glottal fry (β = 0.946, 2.58 ×, 2.07–3.19×), loud voicing (β = 0.276, 1.32 ×, 1.18–1.47×), and modal register (β = 0.982, 2.67 ×, 2.35–3.05×). Beyond forced whispering, loud modal register exceeded modal register (β = 0.704, 2.02 ×, 1.81–2.27×) and the falsetto register exceeded glottal fry (β = 0.255, 1.29 ×, 1.05–1.59×); the glottal-fry vs modal contrast was uncertain (HPD overlapped zero).

**1–2.5 μm size bin.** At the 1–2.5 μm particle size range, forced whispering no longer exhibited robustly higher particle concentrations than other tasks. The loud modal register produced the highest concentrations, exceeding/h/-onset (β ≈ 0.204; ratio ≈ 1.23 ×, 95% HPD 1.09–1.36×), the falsetto register (β ≈ 0.579; 1.78 ×, 1.50–2.09×), glottal fry (β ≈ 0.424; 1.53 ×, 1.22–1.94×), the modal register (β = 0.469; 1.60 ×, 1.39–1.82×), and forced whispering (β = 0.969; 2.63 ×, 2.27–3.10×). The/h/-onset condition was higher than the modal register (β = 0.265; 1.30 ×, 1.14–1.51×), falsetto (β = 0.375; 1.45 ×, 1.22–1.71×), and forced whispering (β = 0.764; 2.15 ×, 1.83–2.52×), but its difference from glottal fry was uncertain (HPD overlapped zero). Relative to forced whispering, the modal register (β = 0.498; 1.65 ×, 1.36–1.98×), the falsetto register (β = 0.390; 1.48 ×, 1.22–1.80×), and glottal fry (β = 0.545; 1.72 ×, 1.33–2.25×) were all higher. Overall, at 1–2.5 μm, the loud modal register dominates and forced whispering is lowest, with/h/-onset, modal register, falsetto register, and glottal fry forming an intermediate tier.

**2.5–5 μm size bin.** The/h/-onset condition (/h/ then vowel) was higher than forced whisper (β = 0.681; ratio ≈ 1.98 ×, 95% HPD 1.14–3.55×), the modal register (β = 0.351; 1.42 ×, 1.08–1.84×), and the falsetto register (β = 0.462; 1.59 ×, 1.03–2.52×). Differences between/h/-onset and loud modal speech were uncertain (β = 0.165; 1.18 ×, 0.96–1.49×), as were all other pairwise contrasts among tasks (HPDs overlapped zero).

**5–10 μm size bin.** No pairwise task differences were credibly detected: all 95% HPD intervals included zero. Taken together, the 5–10 μm bin shows no robust separation by task, consistent with the broader pattern that mid-size particles yield fewer detectable differences

**10–20 μm size bin.** Forced whispering produced the highest coarse-mode concentrations. Relative to forced whisper, loud modal register was lower (Whisper − Loud: β = 0.294; ratio ≈ 1.34 ×, 95% HPD 1.17–1.55×), modal register was lower (β = 0.555; 1.74 ×, 1.44–2.12×),/h/-onset was lower (β = 0.461; 1.59 ×, 1.36–1.84×), falsetto register was lower (β = 0.537; 1.71 ×, 1.46–1.99×), and glottal fry was lower (β = 0.481; 1.62 ×, 1.12–2.29×). Among the non-whisper tasks, loud modal register exceeded modal register (β = 0.259; 1.30 ×, 1.07–1.56×), and loud exceeded/h/-onset (β = 0.164; 1.18 ×, 1.02–1.37×); loud modal register also exceeded falsetto register (β = 0.241; 1.27 ×, 1.08–1.47×). Other pairwise differences in this bin were uncertain (HPDs overlapped zero). These results indicate that forced whispering is unique in its high

emissions at both the smallest and largest size ranges, while other tasks exhibit a consistent decline in aerosol generation as particle size increases.

## Comparison of phonation mechanisms by particle concentration

A Bayesian regression model was used to evaluate the effects of $CO_2$ range and normalized glottal gap on particle concentration, with both predictors standardized to allow for relative interpretation of their effects (**Fig 3**). Predictors of overall emission (APS totals). Both predictors were positively associated with total aerosol concentration, with normalized glottal gap (NGG) the dominant effect. On the log scale, NGG's coefficient was 0.49 (95% CrI 0.44–0.53), implying a ~ 1.63 × increase in expected concentration per 1-SD increase in NGG (ratio = exp(β) = 1.64, 95% CrI 1.55–1.70). $CO_2$ range was smaller but credibly positive: β = 0.16 (95% CrI 0.11–0.20), a ~ 1.17 × change per 1-SD (95% CrI 1.12–1.22). Between-participant variability was substantial (random-intercept SD = 0.67 on the log scale; ≈ 1.95 × multiplicative SD across subjects), and residual dispersion was σ = 0.79. Taken together, NGG explains markedly more variation in emissions than ventilatory output ($CO_2$), with non-overlapping 95% credible intervals and a much larger multiplicative effect. **Fig 4** reflects differences in these variables by task.

## Discussion

This study investigated how different types of phonation influence aerosol generation, focusing on two key physiological factors: $CO_2$ range and normalized maximum glottal opening (NGG). Phonation types that had greater ventilatory output and a larger open-phase glottal gap (e.g., forced whispering, loud modal register) produced the highest aerosol concentrations; types with less ventilatory output and smaller open-phase glottal gap (e.g., glottal fry and modal register) produced the lowest. Smaller aerosol particles (0.1–1 µm) were more prevalent across all vocal tasks, highlighting their potential for airborne transmission. However, forced whispering uniquely produced high emissions at both the smallest and largest particle sizes. NGG showed the strongest positive association with aerosol concentration. Tasks characterized by wider glottal configurations (e.g., whispering,/h/-onset) tended to exhibit higher concentrations. $CO_2$ range was also positively associated with concentration, consistent with greater ventilatory output in louder or whispered productions.

Although whispering is often perceived as a quieter mode of voice production, the forced, high intensity whispering elicited in our task was associated with high aerosol concentrations. Whisper tasks can involve a wider glottal configuration and substantial ventilatory output, which is consistent with the observed levels; however, mechanisms (e.g., flow or turbulence) were not measured and are not inferred. Thus, our findings highlight that not all forms of whispering should be assumed to entail lower aerosol emission. Looking at **Fig 4**, whispering had the highest $CO_2$ range across tasks and the largest NGG. Therefore, the likely reason for the higher concentration of particles was the combination of both the higher

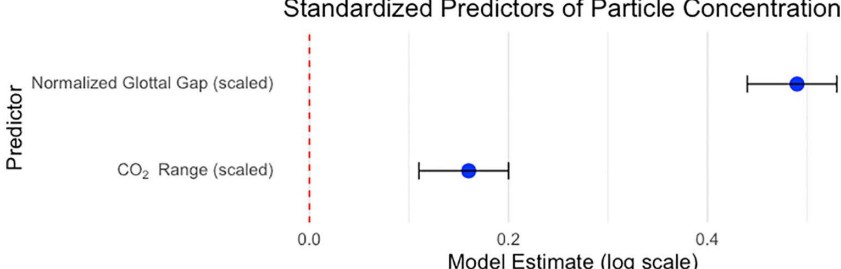

**Fig 3. Forest plot illustrating the standardized regression coefficients for predictors of particle concentration.** Point estimates are shown as blue dots, with black horizontal lines representing 95% highest posterior density intervals. The red dashed vertical line at zero indicates no effect. Predictors include $CO_2$ Range (scaled) and Normalized Glottal Gap (scaled), all standardized for relative interpretation.

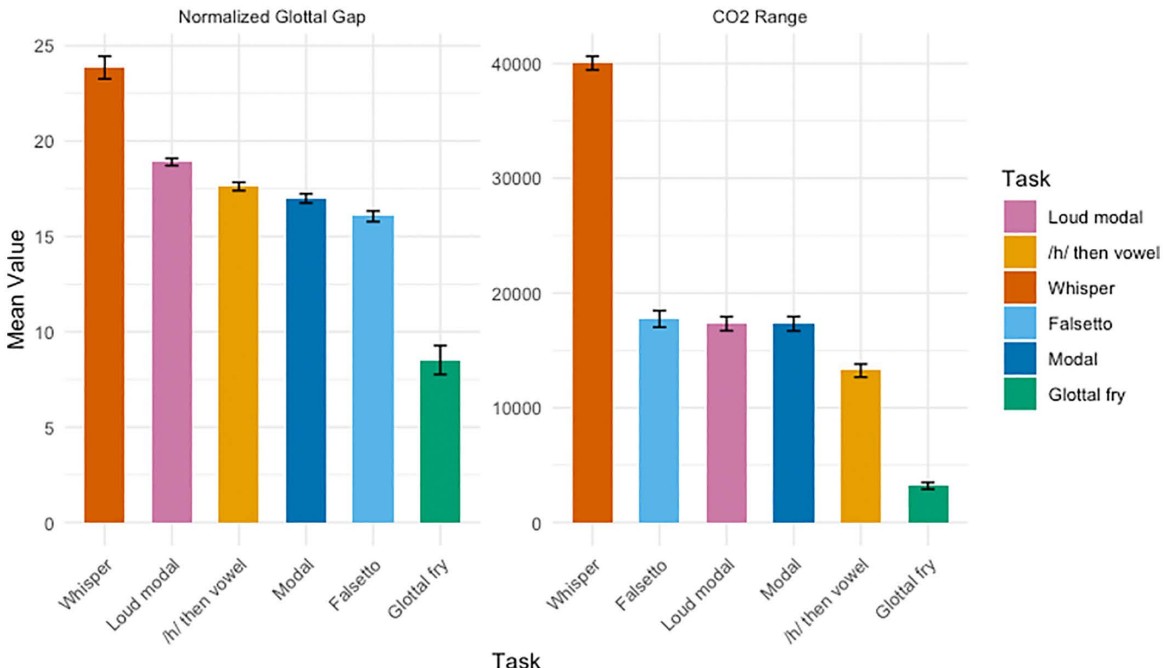

**Fig 4. Mean differences in vocal tasks for Normalized Glottal Gap and CO$_2$ Range with standard error bars.** Each bar represents the mean value for a specific vocal task, with tasks ordered from largest to smallest mean within each facet. Error bars indicate ±1 standard error. The x-axis lists the vocal tasks, and the y-axis shows the mean value for each variable.

airflow and the more open glottal posture. This finding is consistent with recent research from [11,12] showing higher aerosol concentration from whispered speech. An unexpected finding was that whispering showed a bimodal particle-size distribution, with elevated concentrations at both 0.1–1 μm and 10–20 μm. In contrast, the other phonation types exhibited the usual pattern of progressively lower concentrations at larger sizes. Because airflow and pressure were not measured, we do not infer mechanisms; we simply note that whisper tasks typically involve a wider glottal configuration and can entail substantial ventilatory output. The bimodal pattern should be verified in larger samples with direct aerodynamics to determine whether it generalizes.

Also consistent with prior research, louder phonation resulted in greater aerosol concentration, likely driven by respiratory forces and higher amplitude of vocal fold vibration. Loud voicing typically requires higher subglottal pressure as well as stronger expiratory force and greater airflows. Loud voicing particularly generated higher emissions at the smallest size range, with less pronounced differences across larger particle sizes. This finding is consistent with [8] showing higher particle concentrations for louder voicing. Because airflows were not recorded, we avoid mechanistic claims and instead report associations with ventilatory output (CO$_2$ range).

Glottal fry produced the smallest concentration of particles. This finding is consistent with the two physiological mechanisms studied: glottal fry had the smallest glottal gap and the lowest CO$_2$ range. This finding further reinforces the importance of these two variables, particularly airflow and glottal separation, in aerosol generation.

A critical aspect of this study was controlling for CO$_2$ range, ensuring that variations in respiratory effort did not confound differences in aerosol concentration across phonation tasks. CO$_2$ range served as a proxy for ventilatory output, with larger CO$_2$ fluctuations indicating higher ventilatory output and possibly higher airflow velocity. By accounting for CO$_2$ range in the statistical model, we isolated the effect of phonatory mechanisms (i.e., glottal positioning) on aerosol

generation. The high aerosol concentration across vocal tasks, despite controlling for $CO_2$ range, underscores the role of airflow and glottal configuration in aerosol production. However, the finding that whispering, a voiceless production task, produced such high concentration suggests that vocal fold vibration is not required for high aerosol output. Instead, airflow dynamics and glottal aperture size appear to be the primary drivers of aerosol generation, aligning with [10] who proposed that turbulent airflow, and not phonatory periodicity, enhances aerosolization. NGG and $CO_2$ range index related aspects of voice production (i.e., laryngeal aperture and ventilatory output) that are physiologically coupled. Our mixed-effects models, therefore, interpret each coefficient conditional on the other rather than as independent mechanistic effects. The positive associations for both predictors indicate that, within our tasks, larger glottal opening and greater ventilatory output are each associated with higher aerosol counts. Because we did not collect direct subglottal pressure or airflow, we refrain from attributing these effects to a specific turbulence mechanism.

These findings have important implications for airborne disease transmission. As also noted in prior work [11,12], whispering is not necessarily a safer alternative for reducing aerosol emissions, particularly when produced as a forced or "stage" whisper, as in our task. Our study did not examine confidential, low intensity whispering, and it remains possible that such productions, with airflow closer to modal voice, may yield lower particle counts. Still, the high concentration of small particles (0.1–1 μm) observed in forced whispering and loud voicing suggests that these tasks may contribute meaningfully to airborne disease spread, as smaller particles remain suspended in the air longer and travel further. In addition, the production of larger particles (10–20 μm) during whispering raises concerns about fomite transmission, since such particles can settle on surfaces.

While compelling, the results of this study had several limitations. First, the small sample size limits generalizability, and future studies should include larger, more diverse participant groups. Second, only sustained vowel production was used to attempt to isolate aerosol generation at the laryngeal level (and not from movement in the oral cavity). Future studies should examine aerosol production in naturalistic conversational speech, as individuals vary their speech intensity, airflow, and articulation dynamically. For example, projected speech in noisy environments (e.g., classrooms, restaurants) may increase aerosol emissions, while a modal register in one-on-one settings may minimize risk. Investigating these real-world speech variations would help refine public health guidelines for airborne disease mitigation in different social contexts. Third, there is evidence that the APS is less efficient at measuring liquid particles 10 μm or higher [36], however, it is not well understood if particles emitted from the respiratory tract are liquid or solid. We do know that they rapidly undergo the evaporation of some water and particle size shrinkage [37]. Fourth, NGG was obtained on sustained/i/ during laryngoscopy, whereas aerosol measurements were acquired earlier during separate vocal tasks. While NGG is known to vary within and across productions (e.g., with onset, repetition, pitch, or intensity), here it is used as a participant-level characteristic to capture general tendencies rather than a task-specific state. Although the association between NGG and aerosol output suggests that glottal opening contributes to emission, task-to-task variation in posture, pitch, and supraglottic configuration may modulate instantaneous glottal area. Future work should pair simultaneous flow/pressure and within-task glottal kinematics to test efficiency-based mechanisms directly. Finally, because the whisper task was a forced, high-intensity whisper, the results pertain to whispering performed at relatively high expiratory drive and perceived loudness and should not be generalized to quiet conversational whispers. The elevated emissions we observed for forced whisper likely reflect the demands of producing audibility without voicing; however, without concurrent SPL and airflow, we refrain from mechanistic claims. Future studies should systematically vary whisper intensity, record SPL and oral airflow concurrently, and compare quiet versus stage whispers to quantify any intensity-dependent change in particle output.

## Conclusion

This study highlights the significant role of increased expiratory airflow and glottal configuration in aerosol generation during phonation, with forced whispering and loud modal register producing the highest particle concentrations and glottal fry generating the least. By controlling for $CO_2$ range, we isolated the effects of phonatory mechanisms and found that

vocal fold vibration alone is not the primary driver of aerosol production; instead, increased airflow and a more open glottal posture contribute to higher emissions. Notably, forced whispering produced a bimodal distribution of aerosol particles, with elevated concentrations at both the smallest and largest particle sizes, emphasizing its potential role in both airborne and fomite transmission. These findings reinforce the importance of considering speech-driven aerosolization in public health recommendations, particularly in high-density indoor environments where airborne disease transmission is a concern. Future research should explore conversational speech patterns and real-world speech variability to refine strategies for minimizing aerosol emissions in different communicative settings.

## Acknowledgments

Thank you to Dr. Marie Jetté at the University of Colorado Anschutz Medical Campus Department of Otolaryngology for performing the laryngoscopy testing.

## Author contributions

**Conceptualization:** Allison Hilger, Shelly Miller.

**Data curation:** Allison Hilger, Corey Murphey.

**Formal analysis:** Allison Hilger.

**Funding acquisition:** Allison Hilger, Shelly Miller.

**Investigation:** Allison Hilger, Tehya Stockman.

**Methodology:** Allison Hilger, Tehya Stockman, Jacqueline McCurdy.

**Project administration:** Allison Hilger.

**Resources:** Allison Hilger, Shelly Miller.

**Software:** Corey Murphey.

**Supervision:** Shelly Miller.

**Visualization:** Allison Hilger.

**Writing – original draft:** Allison Hilger.

**Writing – review & editing:** Allison Hilger, Tehya Stockman, Corey Murphey, Jacqueline McCurdy, Shelly Miller.

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
