## [Decision Letter · Decision Letter 0]

22 Jul 2025

Dear Dr. Hilger,

1. Use of CO2-range as a “respiratory-effort” proxy. Please define precisely what you mean by “effort” (air-flow? sub-glottal pressure? lung-volume change), as noted by R2. Cite evidence that exhaled-CO2 dynamics capture that construct, or add a direct airflow / pressure measure; otherwise restrict conclusions to speculation.

2. Clarify glottal-gap measurement and reliability. Please explain that gap was recorded in a separate laryngoscopy session on /i/ and discuss its applicability to later aerosol tasks, as noted by R2. Further provide intra-/inter-rater reliability statistics and avoid calling a larger gap “phonatory efficiency” unless backed by flow/pressure data.

3. Improve acoustic predictors and control for loudness. Re-extract HNR (or a more suitable periodicity measure) from the head-mounted microphone used during aerosol capture, as noted by R2. Report sound-pressure levels per condition and separate HNR’s role as periodicity vs. turbulence indicator.

4. Statistical methods. Please consider re-fitting models with a canonical log link and include participant random effects to reflect repeated measures; a canonical log link is often preferred for lognormal means as the distribution is bounded below zero and skews right, whereas the identity assumes an unbounded distribution. 

We look forward to receiving your revised manuscript.

Kind regards,

Steven R Livingstone

Academic Editor

PLOS ONE

Journal Requirements:

This study was made possible with funding from and partnership with the National Federation of State High School Associations and the College Band Directors National Association, as well as the Research and Innovation Office at the University of Colorado Boulder.

NO authors have competing interests

4. Thank you for uploading your study's underlying data set. Unfortunately, the repository you have noted in your Data Availability statement does not qualify as an acceptable data repository according to PLOS's standards.

Reviewers' comments:

Reviewer's Responses to Questions

**Comments to the Author**

1. Is the manuscript technically sound, and do the data support the conclusions?

Reviewer #1: Yes

Reviewer #2: Partly

2. Has the statistical analysis been performed appropriately and rigorously?

Reviewer #1: Yes

Reviewer #2: I Don't Know

3. Have the authors made all data underlying the findings in their manuscript fully available?

Reviewer #1: Yes

Reviewer #2: Yes

4. Is the manuscript presented in an intelligible fashion and written in standard English?

Reviewer #1: Yes

Reviewer #2: Yes

Reviewer #1: This is an automated report for PONE-D-25-11745. This report was solicited by the PLOS One editorial team and provided by ScreenIT.

ScreenIT is an independent group of scientists developing automated tools that analyze academic papers. A set of automated tools screened your submitted manuscript and provided the report below. Each tool was created by your academic colleagues with the goal of helping authors. The tools look for factors that are important for transparency, rigor and reproducibility, and we hope that the report might help you to improve reporting in your manuscript. Within the report you will find links to more information about the items that the tools check. These links include helpful papers, websites, or videos that explain why the item is important. While our screening tools aim to improve and maintain quality standards they may, on occasion, miss nuances specific to your study type or flag something incorrectly. Each tool has limitations that are described on the ScreenIT website. The tools screen the main file for the paper; they are not able to screen supplements stored in separate files. Please note that the Academic Editor had access to these comments while making a decision on your manuscript. The Academic Editor may ask that issues flagged in this report be addressed. If you would like to provide feedback on the ScreenIT tool, please email the team at ScreenIt@bih-charite.de. If you have questions or concerns about the review process, please contact the PLOS One office at plosone@plos.org.

Reviewer #2: The authors are commended for their study aiming to elucidate the actual voice-production related factors of aerosol generation. It sheds quite a bit of light on these mechanisms, and increases our knowledge of the topic considerably.

As I am a voice specialist, and not an aerosol scientist, I will confine my review to the aspects of voice production which have been discussed in the text, and will leave it to other reviewers to address the aerosol measurement methods used.

My main concerns arise over your use and discussion around your predictors/confounders, CO2 Range, NGG and HNR:

CO2 Range

* I have concerns over the reliance on this measure as a proxy for airflow (222-223 and elsewhere). Can you explain HOW this is a robust enough indicator of 'respiratory effort and lung volume', 'amount of air expelled', supporting this with literature? This seems to be a very, very important point as your analysis hinges on this, and how you have controlled for this (essentially, controlling for airflow - which would seem to be very important to really elucidate the mechanisms of aerosol production).

* 101-102: Please clarify what 'respiratory effort' means - is this regarding muscle activation, minute ventilation, respiratory rate? Also, do you mean 'lung volume EXPENDED'? Because speaking of 'lung volume' is usually understood to mean at what relative volume (TLC, RLV, etc).

* Line 443: Again, you are referring to 'higher lung volume' and it is unclear whether you mean a higher lung volume initiation (i.e. at 70% of TLC) or you mean a higher lung volume expenditure (i.e. 30% of VC)

Normalised Glottic Gap (NGG)

* It is important that you specify that you are talking about a normalised glottic gap at the point of MAXIMUM excursion of the vocal folds in their oscillation. Glottic gap is most often, to my mind, used to indicate the gap at the point of maximum closure - indicating glottic incompetence.

* It is important to acknowledge somewhere in the text that the NGG that you have measured was during laryngoscopy, and not at any other point. It is entirely possible that the true values of glottic gap were different during the aerosol measuring procedure. Also, you have only measured NGG during an /i/ vowel, and have not done this for the other vowels you have elicited.

* Getting the mean of three cycles is good - however, ideally the calculation of NGG procedure should be carried out by more than one person and an ICC reported for the raters (as it is, essentially, a visual-perceptual judgement task). I suggest, at a minimum, that at least a repeated sample be analysed and then reporting an intra-rater reliability statistic for this.

* Line 401: Here you refer to glottic gap as a proxy for phonatory efficiency, where in other sections you've used it as an indicator for glottic turbulence (of course, they can be related). However, glottic gap as you've defined it does NOT reflect phonatory efficiency. You are using this measure to reflect the maximum opening of the glottis during open phase. Had you defined it as the glottic gap during the closed phase, this would reflect glottic incompetence and would reflect phonatory efficiency. For a true determination of efficiency, you need measures of flow, pressure and loudness. Please reword this section.

*Throughout you have asserted that a higher NGG would lead to turbulence, but this needs a reference from the literature.

HNR

* At times you say that you are relying on HNR to identify the periodicity of the task (249), whilst at others you discuss using this metric as an approximation for turbulence at the glottis (108). Which is it?

* HNR is, of course, correlated with a periodic signal, but you can have a perfectly periodic vocal fold oscillation with a low HNR (think of breathy voice, the incomplete adduction of the vocal folds would lead to increased air turbulence and increased noise content, but the vocal fold oscillation may still be perfectly periodic). You could have measured periodicity with a jitter measure - not perfect, but as you are using sustained vowels, entirely appropriate. Either report true periodicity (using a time-based measure like jitter), or nuance/remove mentions of HNR as a quantification of periodicity (211, 249, 399-402).

* Additionally, a low HNR would indeed indicate a relative lack of harmonic strength - but this does not necessarily have to happen as a result of glottic-level turbulence. You have measured HNR during videolaryngoscopy… which is VERY problematic. You have not reported the background SPL of the room that this was carried out in (especially as you were using stroboscopy, which is noisy), and what the microphone's response curve was, etc. etc. I appreciate that measuring it during laryngoscopy at least gives you the opportunity to correlate the HNR with THAT specific voice production token - but then to use this later in your analysis (especially when the participants were wearing a head-mounted microphone of higher quality - you don't report the microphone used during the laryngoscopy) means that you cannot make claims as to the actual presence of turbulence DURING the aerosol measurement you are making. Could you address this, or could you use HNR from the recordings made during aerosol production?

* The reliance on HNR as a measure of periodicity is also problematic as you have not reported any controlling for loudness. A louder signal would, to my mind, increase the strength of the harmonics, and depending on the phonation type, might increase the proportion of harmonics vs noise in the signal. In fact, you don't report SPL at all (other than asking participants to target 70-75dB). This is a significant confounder and problematic. Can you address this in your discussion at least?

COMMENTS ON THE EXPERIMENTAL TASKS AND THEIR DESCRIPTIONS

* I am struggling to see how an /h/-onset vowel has a higher glottic gap than voiceless whispering. I think what you are talking about is that at the point of the articulation of the /h/ (a consonant), the vocal folds were partially adducted for the /h/ to provide the turbulence we hear as an /h/. This is not a glottic gap (as it is usually defined). If the /h/ is followed by a vowel, then the glottic gap would be measured during the vowel. I struggle to see how this could possibly be larger than for whispering, when the vocal folds are held open throughout the production of the task. Moreover, the /h/ articulation actually occurs for only milliseconds - and is then followed by (I presume) a normal voice quality vowel (? you have not specified - were you aiming at a breathy phonation quality?). This is likely why the /h/ onset task had a CO2 range fairly similar to the other vowel productions. However, if you were asking for a BREATHY phonation quality in this h-onset task, then this needs to be stated more clearly. This needs clarification throughout the text.

* Your descriptions of modal voice as having a 'fully closed' glottis (116-117) and glottal fry as having 'partially closed' glottis are wrong (also 129-131). I think you are aiming at closure quotient? In this, you would mean how long, relative to the total glottal cycle, the vocal folds are fully closed? This would be higher in fry, lower in modal, and possibly lower still in falsetto. But, all three voice conditions likely achieve full closure. Please reword this to reflect higher/lower closure quotient, predominent open phase/closed phase, etc.

* At times you refer to voicing (151 - 'loud voicing') and everywhere else - as 'speech' (all the figures, 216, 305) you refer to this as . For example, you refer to the whispered vowel condition as 'whispered speech' (218), and modal phonation as 'modal speech'. What you've described is NOT speech, it is loud/whispered vowels/phonation. Speech would involve articulation, and none of your tasks include articulation (save for the /h/ in the /h/-onset task). This was confusing as at several points I wanted to point out that a forced whispered SPEECH/or LOUD SPEECH would involve articulation and this would be a prime candidate for larger particles. I suggest renaming your tasks as:

Sustained vowels in the following registers/conditions:

* Glottal fry (or pulse register)

* Modal register

* Falsetto register

* H-onset (or 'easy onset') [or if it's breathy phonation, you need to say so]

* Loud modal register (comfortable +10dB - or do you mean 10dB louder than the others, i.e. 80-85dB? Please clarify)

* Voiceless (Force Whispered)

DISCUSSION

* Have you considered that NGG and the CO2 range (as a proxy for airflow) are actually interrelated, as a higher NGG would lead to higher transglottic airflows? How have you controlled for this? You do not discuss at any point that what each of these metrics is being utilised as a proxy for (airflow, turbulence, etc) has considerable overlap and really can't be treated as such distinct indicators at all. HNR is affected by loudness, which is impacted by airflow/subglottic pressure, which will impact the CO2 range. HNR will also be impacted by a larger glottic gap. There needs to be a discussion at some point that these predictors, these metrics, are not as independent as you have been presenting them. This should result in a more nuanced discussion of the mechanisms of aerosol generation.

* I do not see that you have successfully differentiated between the role of airflow turbulence and airflow more generally. This is very problematic in my view.

* Any discussion about 'whispering' and its relative safety/risk have to be nuanced with the fact that what you have elicited from your participants was a FORCED whisper (perhaps a 'stage whisper')... a whisper which was targeted to be as loud as the rest of the tasks (bar the 'loud phonation' task at +10db). I think that you basically touch on this in your discussion around lines 437-439, but it is worth noting that not every whisper is equal!

Minor -

Line 79 - in the oral cavity during speech, we have the movement and approximation of ALL the articulators (not simply the opening and closing of the mouth) - generating many opportunities for fluid film burst, etc

Line 86 - Loud speech and singing MAY be initiated by higher lung volumes (to capitalise on higher recoil pressures) but this is NOT a certainty, as loudness can be achieved by lower lung volumes accompanied by higher expiratory and adductory forces. Please rephrase/remove.

Line 90 - Somewhat proves the point of the above (and I presume would be due to higher glottic-level tension/the supraglottic compression we frequently see for people who phonate into low lung volumes) - If both high lung volumes and low lung volumes increases aerosol generation - what point are you making? Obviously breath-holding will decrease aerosol generation as there is no airflow. I suggest either combining talk of high lung volumes with low lung volumes - or leave this out.

Line 152: The vowel in 'sun' is incorrect. It should be /^/

184: If you're going to talk about the vocal processes, you might as well refer to the 'anterior commissure'.

**Do you want your identity to be public for this peer review?** For information about this choice, including consent withdrawal, please see our Privacy Policy

Reviewer #1: No

Reviewer #2: No

---

## [Author Response · Author response to Decision Letter 1]

3 Oct 2025

Dear Dr. Livingstone,

We thank the editor and reviewers for their thoughtful and constructive feedback. We have revised the manuscript substantially to address all major and minor points. Below, we provide a point-by-point response. Reviewer comments are in bold, followed by our responses in regular text. We have made the revisions to this manuscript using red ink.

Thank you,

Allison Hilger, Tehya Stockman, Corey Murphey, Jacqueline McCurdy, and Shelly Miller

Comments from the Editor:

Please consider the reviewer's comments when making your revisions. Below is a short list of major issues identified.

1. Use of CO2-range as a “respiratory-effort” proxy. Please define precisely what you mean by “effort” (air-flow? sub-glottal pressure? lung-volume change), as noted by R2. Cite evidence that exhaled-CO2 dynamics capture that construct, or add a direct airflow / pressure measure; otherwise restrict conclusions to speculation.

a. Response: We have clarified that “respiratory effort” in this study refers specifically to ventilatory output (exhaled airflow volume and velocity), not subglottal pressure or muscular activation. To support the validity of exhaled CO₂ dynamics as a ventilation proxy, we now cite Bhavani-Shankar & Philip (2000) and Russell et al. (1998). We also acknowledge in the revised Discussion that CO₂ range is not a direct measurement of subglottal pressure or airflow mechanics at the glottis, and we now restrict our interpretation to its role as a ventilation control variable. We have also revised the title of the manuscript to reflect this change: “The Influence of Glottal and Respiratory Factors on Aerosol Emission During Phonation”

2. Clarify glottal-gap measurement and reliability. Please explain that gap was recorded in a separate laryngoscopy session on /i/ and discuss its applicability to later aerosol tasks, as noted by R2. Further provide intra-/inter-rater reliability statistics and avoid calling a larger gap “phonatory efficiency” unless backed by flow/pressure data.

a. Response: Thank you for this helpful point. We have revised the manuscript accordingly. We now state explicitly that normalized glottal gap (NGG) was measured during a separate videolaryngoscopy session on sustained /i/ at comfortable pitch and loudness, not during the aerosol trials (stated in the methods and the limitations). We report intra-rater reliability and manual-versus-algorithm agreement. We also removed any references to “phonatory efficiency”: in this study, a larger NGG is interpreted only as greater maximum glottal opening under the measured posture; we do not infer efficiency without concurrent flow, subglottal pressure, and SPL. Finally, we clarify in the Discussion that NGG serves as a between-participant anatomical/physiological marker rather than a within-task state measure; its applicability across different phonation types is therefore limited, and we frame NGG–aerosol findings as associations rather than mechanistic proof.

3. Improve acoustic predictors and control for loudness. Re-extract HNR (or a more suitable periodicity measure) from the head-mounted microphone used during aerosol capture, as noted by R2. Report sound-pressure levels per condition and separate HNR’s role as periodicity vs. turbulence indicator.

a. Response: We agree that any acoustic predictor should be derived from the same recording chain used during aerosol capture and interpreted separately from loudness. In our dataset, however, no calibrated head-mounted audio was archived for the aerosol trials; the only HNR we originally computed came from laryngoscopy audio, which we now recognize is contaminated by endoscopy/strobe noise and does not reflect the acoustic field during APS sampling. To avoid misleading inferences, we removed HNR from all models and text and do not make claims about periodicity.

4. Statistical methods. Please consider re-fitting models with a canonical log link and include participant random effects to reflect repeated measures; a canonical log link is often preferred for lognormal means as the distribution is bounded below zero and skews right, whereas the identity assumes an unbounded distribution.

a. Response: We note the suggestion to use a canonical log link for lognormal outcomes. In brms, the lognormal family is parameterized on the log scale by default (with identity link predicting log-transformed means), so our models already conform to this specification. We have confirmed this parameterization and clarified it in the Methods. We also re-fit all inferential models with participant random intercepts to capture within-subject dependence:

a) Response: The revised manuscript has been updated for the formatting requirements for the journal.

b) Thank you for stating the following financial disclosure:

This study was made possible with funding from and partnership with the National Federation of State High School Associations and the College Band Directors National Association, as well as the Research and Innovation Office at the University of Colorado Boulder. Please state what role the funders took in the study. If the funders had no role, please state: "The funders had no role in study design, data collection and analysis, decision to publish, or preparation of the manuscript."

i) Response: This has been revised.

c) Thank you for stating the following in the Competing Interests section:

NO authors have competing interests

i) Response: This has been revised.

d) Thank you for uploading your study's underlying data set. Unfortunately, the repository you have noted in your Data Availability statement does not qualify as an acceptable data repository according to PLOS's standards.

i) Response: The data have been uploaded in fighsare.

i) Response: Thank you. The citations have been evaluated and included as relevant.

Reviewers' comments:

5) Comments to the Author

6) Reviewer #1: This is an automated report for PONE-D-25-11745. This report was solicited by the PLOS One editorial team and provided by ScreenIT.

ScreenIT is an independent group of scientists developing automated tools that analyze academic papers. A set of automated tools screened your submitted manuscript and provided the report below. Each tool was created by your academic colleagues with the goal of helping authors. The tools look for factors that are important for transparency, rigor and reproducibility, and we hope that the report might help you to improve reporting in your manuscript. Within the report you will find links to more information about the items that the tools check. These links include helpful papers, websites, or videos that explain why the item is important. While our screening tools aim to improve and maintain quality standards they may, on occasion, miss nuances specific to your study type or flag something incorrectly. Each tool has limitations that are described on the ScreenIT website. The tools screen the main file for the paper; they are not able to screen supplements stored in separate files. Please note that the Academic Editor had access to these comments while making a decision on your manuscript. The Academic Editor may ask that issues flagged in this report be addressed. If you would like to provide feedback on the ScreenIT tool, please email the team at ScreenIt@bih-charite.de. If you have questions or concerns about the review process, please contact the PLOS One office at plosone@plos.org.

Reviewer #2: The authors are commended for their study aiming to elucidate the actual voice-production related factors of aerosol generation. It sheds quite a bit of light on these mechanisms, and increases our knowledge of the topic considerably.

Response: Thank you for this positive review of our paper. Your comments have greatly improved the rigor of the study.

7) As I am a voice specialist, and not an aerosol scientist, I will confine my review to the aspects of voice production which have been discussed in the text, and will leave it to other reviewers to address the aerosol measurement methods used.

My main concerns arise over your use and discussion around your predictors/confounders, CO2 Range, NGG and HNR:

CO2 Range

* I have concerns over the reliance on this measure as a proxy for airflow (222-223 and elsewhere). Can you explain HOW this is a robust enough indicator of 'respiratory effort and lung volume', 'amount of air expelled', supporting this with literature? This seems to be a very, very important point as your analysis hinges on this, and how you have controlled for this (essentially, controlling for airflow - which would seem to be very important to really elucidate the mechanisms of aerosol production).

* 101-102: Please clarify what 'respiratory effort' means - is this regarding muscle activation, minute ventilation, respiratory rate? Also, do you mean 'lung volume EXPENDED'? Because speaking of 'lung volume' is usually understood to mean at what relative volume (TLC, RLV, etc).

* Line 443: Again, you are referring to 'higher lung volume' and it is unclear whether you mean a higher lung volume initiation (i.e. at 70% of TLC) or you mean a higher lung volume expenditure (i.e. 30% of VC)

Response: We also agree that the terminology was too loose. Throughout the manuscript, we have replaced “respiratory effort” with “ventilatory output.” To support the validity of exhaled CO₂ dynamics as a ventilation proxy, we now cite Bhavani-Shankar & Philip (2000) and Russell et al. (1998). We explicitly note that CO₂ range is not a direct measure of airflow, subglottal pressure, muscle activation, or absolute lung volume. We also removed ambiguous references to “higher lung volume” and now distinguish lung-volume initiation (starting phonation nearer total lung capacity) from lung-volume expenditure (fraction of vital capacity exhaled during the task). Because we did not measure lung volumes, we refrain from claims about either and instead refer to higher ventilatory output when relevant, except when we are citing other studies that do measure lung volume. These clarifications are added to Methods and the affected sentences in the Introduction/Discussion have been revised accordingly.

Normalised Glottic Gap (NGG)

* It is important that you specify that you are talking about a normalised glottic gap at the point of MAXIMUM excursion of the vocal folds in their oscillation. Glottic gap is most often, to my mind, used to indicate the gap at the point of maximum closure - indicating glottic incompetence.

* It is important to acknowledge somewhere in the text that the NGG that you have measured was during laryngoscopy, and not at any other point. It is entirely possible that the true values of glottic gap were different during the aerosol measuring procedure. Also, you have only measured NGG during an /i/ vowel, and have not done this for the other vowels you have elicited.

* Getting the mean of three cycles is good - however, ideally the calculation of NGG procedure should be carried out by more than one person and an ICC reported for the raters (as it is, essentially, a visual-perceptual judgement task). I suggest, at a minimum, that at least a repeated sample be analysed and then reporting an intra-rater reliability statistic for this.

* Line 401: Here you refer to glottic gap as a proxy for phonatory efficiency, where in other sections you've used it as an indicator for glottic turbulence (of course, they can be related). However, glottic gap as you've defined it does NOT reflect phonatory efficiency. You are using this measure to reflect the maximum opening of the glottis during open phase. Had you defined it as the glottic gap during the closed phase, this would reflect glottic incompetence and would reflect phonatory efficiency. For a true determination of efficiency, you need measures of flow, pressure and loudness. Please reword this section.

*Throughout you have asserted that a higher NGG would lead to turbulence, but this needs a reference from the literature.

Response: We agree and have modified the methods and discussion in light of your comments. We now state explicitly that our NGG metric reflects the maximum glottal opening during the open phase (peak abduction within a vibratory cycle). We clarify that NGG was measured during a separate videolaryngoscopy session on sustained /i/ at comfortable pitch and loudness; it was not captured during the APS aerosol trials. We therefore interpret NGG as a between-participant anatomical/physiological characteristic that may co-vary with task production, and we explicitly note that the glottal configuration during aerosol capture could differ from the laryngoscopy token (and from other vowels). We also have calculated reliability. Intra-rater consistency on a random subset of 15 tokens was excellent: ICC(3,1) = 0.924, 95% CI 0.853–0.969. Approximately 40% of images were pre-segmented with a Canny edge-detection workflow and then verified by the rater; the remainder were traced manually. Finally, we removed all references to NGG as “phonatory efficiency.” As the reviewer notes, efficiency requires concurrent flow, pressure, and SPL; our NGG is strictly a geometric descriptor of maximum open-phase area.

HNR

* At times you say that you are relying on HNR to identify the periodicity of the task (249), whilst at others you discuss using this metric as an approximation for turbulence at the glottis (108). Which is it?

* HNR is, of course, correlated with a periodic signal, but you can have a perfectly periodic vocal fold oscillation with a low HNR (think of breathy voice, the incomplete adduction of the vocal folds would lead to increased air turbulence and increased noise content, but the vocal fold oscillation may still be perfectl

---

## [Decision Letter · Decision Letter 1]

29 Oct 2025

Dear Dr. Hilger,

Thank you for submitting your manuscript to PLOS ONE. After careful consideration, we feel that it has merit but does not fully meet PLOS ONE’s publication criteria as it currently stands. Therefore, we invite you to submit a revised version of the manuscript that addresses the points raised during the review process.

We look forward to receiving your revised manuscript.

Kind regards,

Steven R Livingstone

Academic Editor

PLOS ONE

Journal Requirements:

Reviewers' comments:

Reviewer's Responses to Questions

**Comments to the Author**

Reviewer #2: (No Response)

2. Is the manuscript technically sound, and do the data support the conclusions?

Reviewer #2: Yes

3. Has the statistical analysis been performed appropriately and rigorously?

Reviewer #2: Yes

4. Have the authors made all data underlying the findings in their manuscript fully available?

Reviewer #2: Yes

5. Is the manuscript presented in an intelligible fashion and written in standard English?

Reviewer #2: Yes

Reviewer #2: Dear Authors,

This manuscript is MUCH, MUCH improved. Thank you for taking on all of my very, very detailed points! I have one or two tiny little points that still, to my mind, need addressing - but they are minor. Once these are addressed, I think this manuscript needs to be accepted (and I don't need to review it again).

Line 46 - Call me a stickler (which I'm sure you already have, lol) - but I'm not sure you can have 'voiceless phonation' since phonation is typically used to refer to vocal fold vibration. Here, and elsewhere, it might be better to call whispering a 'voiceless sound production', or even 'voiceless production'.

Line 50 - Following on from above, 'sustained sound production'

Line 137 - 'followed by MODAL PHONATION'

Line 159 - I only used /^/ since it's what I can produce on my keypad, but please use the real IPA symbol for this: /ʌ/

DISCUSSION

Line 458 - /h/-onset shouldn't have any wider glottal configuration during phonation (unless it encouraged a slightly breathier phonation style, which you weren't aiming at), and whispering doesn't have phonation. Maybe reword to 'wider glottal configurations' without the 'phonation'. I can see the h-onset NGG during the vowel was actually larger than for modal voice, but within the error bar (Figure 4). It's possible that the task did induce some participants to produce a slightly breathier voice quality? Anyway, you can probably make the same point by saying 'Tasks characterised by wider glottal configurations (e.g. whispering, loud phonation)

Figure 4 - Need to change the 'Loud Speech' labels to 'Loud Modal'

Line 461 - You elicited forced, 'high intensity' (line 162) whispering - so not sure you can really call this a 'quieter production' (especially without even relative SPL values to go on). Also, I'm not familiar with the other literature you cite (11,12) - was their whispering a loud, forced whisper or a whisper one might use to not be overheard. Are your findings really 'contrary to the common assumption that quieter productions carry lower risk'? This needs a bit of rewording

476-486 - I'm not sure that mentioning 'lung volumes' is necessary. Keep line 478 up until, 'therefore requiring initiation of phonation at a higher lung volume', which you can drop. Then you can remove 482-484. I think you can just say that loud voicing typically requires higher SG pressure and stronger expiratory force AND GREATER AIRFLOWS'. Then in 485, 'Because airflows were not recorded...'

498 - 'whispering, a voiceless PRODUCTION task'

509-517 - This does need some nuancing. As mentioned, not all whispers are equal! Line 510 might be better as, 'whispering, ESPECIALLY NOT A FORCED, STAGE WHISPER, is not necessarily a safer alternative...' You have not measured a 'confidential whisper', so, I'm not sure you can say that a clear, low intensity modal register is likely to generate fewer particles. I would be surprised if a genuine, 'confidential' whisper - produced with a similar airflow to modal voice - did not generate fewer particles than quiet modal voice as it would essentially be the same task without vocal fold contact. That said, since Joe Public probably does not know that there's a difference between a quiet whisper and a loud whisper, I think it does bear reporting that a whisper is not necessarily safe. However, I think this paragraph needs some stronger qualifications and caveats. I know you talk about this in the limitations section (line 539), but it needs an extra nod here.

Line 532 - NGG is anything but stable. It will vary over the course of a single vowel from onset to termination, across repetitions of the same vowel, vary at different pitches and intensities (as you say in 535 - instantaneous glottal area which is NGG). That said, intra-participant variation across tasks types are likely to be large enough that the metric is useful for your purposes. I think this is the point you are making?

**Do you want your identity to be public for this peer review?** For information about this choice, including consent withdrawal, please see our Privacy Policy

Reviewer #2: No

---

## [Author Response · Author response to Decision Letter 2]

29 Oct 2025

October 29, 2025

Steven R Livingstone

Academic Editor

PLOS ONE

Manuscript title: Aerosol Emission During Speech: Investigating the Role of Glottal Configuration and Respiratory Effort

Manuscript ID: PONE-D-25-11745

Authors: Hilger et al.

Dear Dr. Livingstone,

We thank the editor and Reviewer One for their thoughtful and constructive feedback. We have revised the manuscript substantially to address all major and minor points. Below, we provide a point-by-point response. Reviewer comments are in bold, followed by our responses in regular text. We have made the revisions to this manuscript using red ink.

I want to note that even though Reviewer One said that they do not need to review the paper again after addressing these comments, we would like to extend our gratitude for their time and expertise. It is evident that they spent quite a bit of time reviewing the paper and providing constructive feedback. They were wonderful to work with.

Thank you,

Allison Hilger, Tehya Stockman, Corey Murphey, Jacqueline McCurdy, and Shelly Miller

Reviewer #2: Dear Authors,

This manuscript is MUCH, MUCH improved. Thank you for taking on all of my very, very detailed points! I have one or two tiny little points that still, to my mind, need addressing - but they are minor. Once these are addressed, I think this manuscript needs to be accepted (and I don't need to review it again).

Line 46 - Call me a stickler (which I'm sure you already have, lol) - but I'm not sure you can have 'voiceless phonation' since phonation is typically used to refer to vocal fold vibration. Here, and elsewhere, it might be better to call whispering a 'voiceless sound production', or even 'voiceless production'.

• Response: Great point. This wording has been revised.

Line 50 - Following on from above, 'sustained sound production'

• Response: This has been revised.

Line 137 - 'followed by MODAL PHONATION'

• Response: This has been revised.

Line 159 - I only used /^/ since it's what I can produce on my keypad, but please use the real IPA symbol for this: /ʌ/

• Response: This has been revised.

DISCUSSION

Line 458 - /h/-onset shouldn't have any wider glottal configuration during phonation (unless it encouraged a slightly breathier phonation style, which you weren't aiming at), and whispering doesn't have phonation. Maybe reword to 'wider glottal configurations' without the 'phonation'. I can see the h-onset NGG during the vowel was actually larger than for modal voice, but within the error bar (Figure 4). It's possible that the task did induce some participants to produce a slightly breathier voice quality? Anyway, you can probably make the same point by saying 'Tasks characterised by wider glottal configurations (e.g. whispering, loud phonation)

• Response: This has been revised to remove reference to phonation. I do agree that this task likely led to a slightly breathier voice quality.

Figure 4 - Need to change the 'Loud Speech' labels to 'Loud Modal'

• Response: Loud speech has been revised to Loud Modal in this figure.

Line 461 - You elicited forced, 'high intensity' (line 162) whispering - so not sure you can really call this a 'quieter production' (especially without even relative SPL values to go on). Also, I'm not familiar with the other literature you cite (11,12) - was their whispering a loud, forced whisper or a whisper one might use to not be overheard. Are your findings really 'contrary to the common assumption that quieter productions carry lower risk'? This needs a bit of rewording

• Response: Great point. We have revised this section to say, “Although whispering is often perceived as a quieter mode of voice production, the forced, high intensity whispering elicited in our task was associated with high aerosol concentrations. Whisper tasks can involve a wider glottal configuration and substantial ventilatory output, which is consistent with the observed levels; however, mechanisms (e.g., flow or turbulence) were not measured and are not inferred. Thus, our findings highlight that not all forms of whispering should be assumed to entail lower aerosol emission.”

476-486 - I'm not sure that mentioning 'lung volumes' is necessary. Keep line 478 up until, 'therefore requiring initiation of phonation at a higher lung volume', which you can drop. Then you can remove 482-484. I think you can just say that loud voicing typically requires higher SG pressure and stronger expiratory force AND GREATER AIRFLOWS'. Then in 485, 'Because airflows were not recorded...'

• Response: Excellent suggestions. This has been revised exactly according to these suggestions.

498 - 'whispering, a voiceless PRODUCTION task'

• Response: This has been revised.

509-517 - This does need some nuancing. As mentioned, not all whispers are equal! Line 510 might be better as, 'whispering, ESPECIALLY NOT A FORCED, STAGE WHISPER, is not necessarily a safer alternative...' You have not measured a 'confidential whisper', so, I'm not sure you can say that a clear, low intensity modal register is likely to generate fewer particles. I would be surprised if a genuine, 'confidential' whisper - produced with a similar airflow to modal voice - did not generate fewer particles than quiet modal voice as it would essentially be the same task without vocal fold contact. That said, since Joe Public probably does not know that there's a difference between a quiet whisper and a loud whisper, I think it does bear reporting that a whisper is not necessarily safe. However, I think this paragraph needs some stronger qualifications and caveats. I know you talk about this in the limitations section (line 539), but it needs an extra nod here.

• Response: Great points. We have added more nuance to this section:

“These findings have important implications for airborne disease transmission. As also noted in prior work (11,12), whispering is not necessarily a safer alternative for reducing aerosol emissions, particularly when produced as a forced or “stage” whisper, as in our task. Our study did not examine confidential, low intensity whispering, and it remains possible that such productions, with airflow closer to modal voice, may yield lower particle counts. Still, the high concentration of small particles (0.1–1 µm) observed in forced whispering and loud voicing suggests that these tasks may contribute meaningfully to airborne disease spread, as smaller particles remain suspended in the air longer and travel further. In addition, the production of larger particles (10–20 µm) during whispering raises concerns about fomite transmission, since such droplets can settle on surfaces.”

Line 532 - NGG is anything but stable. It will vary over the course of a single vowel from onset to termination, across repetitions of the same vowel, vary at different pitches and intensities (as you say in 535 - instantaneous glottal area which is NGG). That said, intra-participant variation across tasks types are likely to be large enough that the metric is useful for your purposes. I think this is the point you are making?

• Response: Great points here as well. We have added more nuance to this section:

“While NGG is known to vary within and across productions (e.g., with onset, repetition, pitch, or intensity), here it is used as a participant-level characteristic to capture general tendencies rather than a task-specific state.”

---

## [Editor Report · Decision Letter 2]

11 Nov 2025

The influence of glottal and respiratory factors on aerosol emission during phonation

PONE-D-25-11745R2

Dear Dr. Hilger,

We’re pleased to inform you that your manuscript has been judged scientifically suitable for publication and will be formally accepted for publication once it meets all outstanding technical requirements.

Kind regards,

Steven R Livingstone

Academic Editor

PLOS ONE
---

## [Editor Report · Acceptance letter]

PONE-D-25-11745R2

PLOS ONE

Dear Dr. Hilger,

I'm pleased to inform you that your manuscript has been deemed suitable for publication in PLOS ONE. Congratulations! Your manuscript is now being handed over to our production team.

Kind regards,

on behalf of

Dr. Steven R Livingstone

Academic Editor

PLOS ONE